# Evaluating and Modeling Attribution
# for Cross-Lingual Question Answering

**Benjamin Muller**[1]♠, **John Wieting**[2], **Jonathan H. Clark**[2],
**Tom Kwiatkowski**[2], **Sebastian Ruder**[2], **Livio Baldini Soares**[2]
**Roee Aharoni**[3], **Jonathan Herzig**[3], **Xinyi Wang**[2]
[1]INRIA Paris, [2]Google DeepMind, [3]Google Research

## Abstract

Trustworthy answer content is abundant in many high-resource languages and is instantly accessible through question answering systems—yet this content can be hard to access for those that do not speak these languages. The leap forward in cross-lingual modeling quality offered by generative language models offers much promise, yet their raw generations often fall short in factuality. To improve trustworthiness in these systems, a promising direction is to *attribute* the answer to a retrieved source, possibly in a content-rich language different from the query. Our work is the first to study attribution for cross-lingual question answering. First, we introduce the XOR-AttriQA dataset to assess the attribution level of a state-of-the-art cross-lingual question answering (QA) system in 5 languages. To our surprise, we find that a substantial portion of the answers is not attributable to any retrieved passages (up to 47% of answers exactly matching a gold reference) despite the system being able to attend directly to the retrieved text. Second, to address this poor attribution level, we experiment with a wide range of attribution detection techniques. We find that Natural Language Inference models and PaLM 2 fine-tuned on a very small amount of attribution data can accurately detect attribution. With these models, we improve the attribution level of a cross-lingual QA system. Overall, we show that current academic generative cross-lingual QA systems have substantial shortcomings in attribution and we build tooling to mitigate these issues.[1]

---

[1]The XOR-AttriQA dataset is available at `https://github.com/google-research/google-research/tree/master/xor_attriqa`. XOR-AttriQA includes approximately 10,000 annotated examples to foster research in the modeling and evaluation of attribution in cross-lingual settings.

[†] Correspondence to {jwieting,jhclark}@google.com.
♠Work done as an intern at Google Research.

## 1 Introduction

Open-Retrieval Question Answering (ORQA) delivers promising performance for information-seeking question answering in about 20 languages (Asai et al., 2021b, 2022; Muller et al., 2022). ORQA models typically consist of a retriever that retrieves documents in a large corpus, followed by a generator that generates a short answer based on the top-ranked documents.

Recent work in ORQA reached a new state of the art by not only retrieving documents in the same language as the query but by also retrieving passages cross-lingually, in additional languages (Asai et al., 2021b). This approach is particularly beneficial for languages with limited online written content (Kornai, 2013; Valentim et al., 2021), potentially allowing users to access information that may not be available in their language.

ORQA models are typically evaluated with string-matching metrics (e.g., Exact-Match) based on extensive collections of question and answer pairs in multiple languages (Clark et al., 2020; Longpre et al., 2021). However, these metrics are limited in three ways. First, they are inherently hard to scale to real-world applications, as they require the collection of gold answers for all the queries. Second, in some cases, the answer can be correct without any overlap with a gold reference (Bulian et al., 2022). Third, short answers are usually not enough to provide a trustworthy answer, and users may prefer to access the underlying source document. To address this last challenge, Bohnet et al. (2022) framed a new task called Attributed Question Answering (AQA). Given a query, AQA consists of predicting a short answer along with a supporting document retrieved from a large corpus (e.g. Wikipedia).

This work is the first study on Attributed Question Answering in the cross-lingual setting.[2] Mea-

---

[2]We refer to a system that generates an answer using text

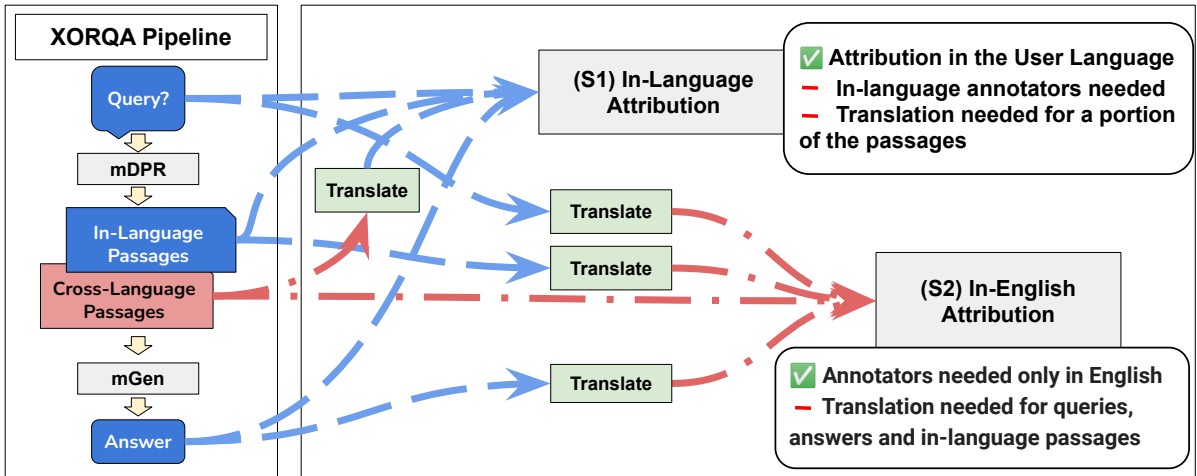

Figure 1: Attribution scenarios for cross-lingual Open-Retrieval Question Answering (XORQA). Given a query in a source language, a retrieval model (MDPR) retrieves source language and cross-lingual documents, which are then used by a generation model, MGEN, to produce a source language answer. For S1, in-language annotators assess attribution directly in the user's language while for S2, annotators validate attribution in English. We collect data for both scenarios in Bengali, Finnish, Japanese, Russian and Telugu.

suring attribution (Rashkin et al., 2021) in the cross-lingual setting is more complex than in the monolingual case. Indeed, in this case, the document supporting the generated answer may be in a language different from the query and answer. Hence, attribution can be defined in various ways depending on the query, document, and answer languages. In this work, we introduce two attribution scenarios, namely (S1) *in-language* attribution for which the attribution is measured in the language of the query, and (S2) *in-English* attribution for which attribution is measured in English. We note that both scenarios may require translating a portion of the documents, the query, or the answer. We illustrate these scenarios in Figure 1.

Based on this framework, we first measure the attribution level of CORA (Asai et al., 2021b), a state-of-the-art cross-lingual ORQA system. We collect data in 5 languages: Bengali, Finnish, Japanese, Russian, and Telugu. To our surprise, a large portion of generated answers was found not attributable to any retrieved passage. For instance, in Japanese, up to 47% of answers exactly matching the gold reference are not attributable. This poor attribution may hurt the trust into our QA systems and limit their deployment.

To improve the attribution level of cross-lingual QA systems, we experiment with a wide range of attribution detection models. We show that PaLM 2

(Anil et al., 2023) outperforms all the other models despite being fine-tuned on a very small sample of our collected data (250 examples). This result shows the potential of using large language models to create state-of-the-art cross-lingual attribution models using very little annotated data, allowing them to be inexpensively created for use in many of the languages of the world. Additionally, we find that for Bengali, Finnish, Japanese and Russian, a T5 model fine-tuned on a large natural language inference corpora reaches very high accuracy compared to the baselines.

Our analysis shows that PaLM 2 can detect more than 86% of *attributed answers* that are not exactly matching the gold reference, showing that it is a useful alternative to exact match evaluation. These answers may be answers that are not captured by the gold references but that are alternative correct answers or they can be answers that are semantically equivalent to the gold reference but that do not overlap with the gold reference (e.g. different units). We discuss these cases in Section 4.2.

In summary, we make the following four contributions: **(i)** Our work is the first to study attribution for question answering in a cross-lingual framework. We define two attribution scenarios, in-language attribution and in-English Attribution and annotate approximately 10,000 examples in five languages; **(ii)** Using this data, we evaluate the attribution of CORA, a state-of-the-art cross-lingual QA system. We show that a large portion (7%-47%,

---

in any language, possibly in a language different from the query, as a cross-lingual QA system.

depending on the language) of the answers are not attributable—neither to in-language passages nor to cross-language ones; **(iii)** We show that PaLM 2 and NLI models can accurately detect attribution in the cross-lingual setting, significantly outperforming all other baselines and reaching above 90% accuracy for all 5 languages. Moreover, our work is the first to approach it with large language models, and we show that with scarce amounts of data (250 examples), they can outperform NLI models trained on millions of examples; and **(iv)** Using our attribution detection model as a reranker, we show that we reach an average of +55% in attribution compared to a model with no reranking.

## 2 Attribution for Cross-Lingual Question Answering

### 2.1 Attribution of Generative Language Models

Generative language models have made impressive progress in the past few years (Radford et al., 2019; Raffel et al., 2020; Brown et al., 2020; Chowdhery et al., 2022). They can now perform most NLP tasks with relatively high accuracy in zero-shot and few-shot settings. However, generating text without reference to human-written trusted sources can be harmful in the real world. Indeed, even the largest models may assign a high probability to false and potentially harmful utterances (Bender et al., 2021; Bommasani et al., 2021; Weidinger et al., 2022). To overcome this challenge, Rashkin et al. (2021) introduced the *Attributable to Identified Sources* (AIS), a human evaluation framework for identifying whether a source document supports the generated text, or in other words, whether the generations can be *attributed* to a given document.

### 2.2 Attributed Question Answering

The need for attribution is particularly vivid for information-seeking use cases. To address this need, Bohnet et al. (2022) defined the *Attributed Question Answering* (AQA) task. Given a query $q$ and a large corpus of text $\mathcal{C}$ (e.g. Wikipedia), AQA consists of predicting an answer $a$ and a passage $p \in \mathcal{C}$ to attribute the predicted answer.

$$(\text{AQA}) \qquad (q, \mathcal{C}) \rightarrow (a, p) \qquad (1)$$

Bohnet et al. (2022) experimented with the AQA task where the questions, answers, and passages were in English. Our work is the first to study this task in a cross-lingual framework.

We build upon previous work that showed that for many languages (e.g., Japanese), cross-lingual QA systems outperform monolingual systems (Asai et al., 2021b; Muller et al., 2022). In this setting, given a query in a language $L$, the goal is to generate an answer $a$ using evidence passages from a multilingual corpus $\mathcal{C}$.

### 2.3 Modeling Attributed Cross-Lingual QA

Asai et al. (2022) showed that the best systems—according to the Exact-Match metric—for question answering in languages other than English are based on a cross-lingual open-retrieval (XORQA) pipeline. We thus model attributed cross-lingual QA with CORA (Asai et al., 2021b), a state-of-the-art XORQA model. Figure 1 (left-panel) illustrates a typical XORQA pipeline.

CORA consists of two components: a multilingual dense retriever (MDPR) which is based on mBERT (Devlin et al., 2019) fine-tuned for dense passage retrieval (Karpukhin et al., 2020), and a multilingual generator (MGEN) based on mT5-Base (Xue et al., 2021) fine-tuned for question answering. Given a query, MDPR ranks all the passages from Wikipedia regardless of their language. In practice, most of the top-ranked passages are either in the same language as the query or in English (we report the language distribution in Table 8 in the Appendix). Then, the top passages are fed to MGEN, which generates the answer. Depending on the languages (and their associated subword tokenization), the average number of passages varies between 5 (for Bengali) and 10 (for Japanese).

CORA is designed to generate a short answer using multiple passages. To use CORA for AQA, we must select a single passage supporting the answer. In this work, we consider the passages that have been ranked highest by MDPR and fed to the generator as our pool of potential attribution passages. We measure and report the attribution level of answers and passages $(a, p)$ by taking the TOP-1-retrieved passage by MDPR as well as ALL the passages retrieved and fed to the generator. Finally, we report in Section 5.3 the attribution level of answers and passages $(a, p)$ *after* reranking the top passages with our NLI-based attribution detection model.

Recall that the selected passage can be in any language but is typically in English or in the query language (cf. Table 8). This leads us to define two attribution evaluation scenarios.

## 2.4 Cross-Lingual QA Attribution Evaluation

We introduce two attribution evaluation scenarios illustrated in Figure 1.

**(S1) In-Language Attribution Evaluation** In this scenario, attribution is assessed in the language of the query, while the query, answer, and passage $(q, a, p)$ are in the same language. From an application perspective, this scenario evaluates directly what a potential user of an attributed QA system would experience by receiving the answer to their question and the attributed source document in their language. As illustrated in Figure 1, this scenario involves automatically translating the portion of the passages retrieved in languages different from the query into the query language.

**(S2) In-English Attribution Evaluation** In this scenario, the query, answer, and passage $(q, a, p)$ are all in English during human annotation; we automatically translate the query and answer into English along with the passages retrieved in languages other than English (cf. Figure 1). We implement this scenario as it favors scalability, since collecting data in English is usually easier than in other languages due to the availability of raters. Moreover, a significant portion of the passages retrieved by cross-lingual QA systems are in English, so assessing attribution directly in English is most straightforward for these passages. For polyglot users, this scenario is also appealing as they may understand English and be interested in accessing the attributed document in English along with the answer in their language.[3]

For both scenarios, translation is performed automatically using the Google Translate API.[4]

**Evaluation Metric** For both scenarios, we collect evaluation data to assess if a predicted answer can be attributed to a retrieved passage. Following Bohnet et al. (2022), we measure the accuracy of a system by counting the proportion of answers with an attributed passage. We refer to this score as AIS.

We note that this evaluation method fundamentally differs from traditional QA system metrics, which are usually based on string-matching methods, e.g., Exact-Match (EM; Rajpurkar et al., 2016; Petroni et al., 2021). Indeed, given a query, answer

---

[3]About 1.5 billion people speak English, as reported by https://www.economist.com/graphic-detail/2019/12/04/where-are-the-worlds-best-english-speakers.

[4]https://cloud.google.com/translate

|  | BN | FI | JA | RU | TE |
|---|---|---|---|---|---|
| *Number of examples collected (# unique queries / # examples)* | | | | | |
| (S1) | 396 / 1560 | 202 / 806 | 146 / 1263 | 186 / 789 | 323 / 1212 |
| (S2) | 305 / 570 | 202 / 812 | 146 / 1262 | 186 / 790 | 228 / 444 |
| *Inter-Annotator Agreement: Agreement with Consensus* | | | | | |
| (S1) | 91.9 | 91.0 | 86.0 | 90.1 | 90.3 |
| (S2) | 87.9 | 92.2 | 86.2 | 94.9 | 98.4 |
| *Disagreement between annotation scenarios* | | | | | |
| S1≠S2 | 6.7 | 10.5 | 5.2 | 7.5 | 4.1 |
| S1≠S2 Tr. | 3.7 | 9.4 | 3.4 | 11.7 | 1.7 |

Table 1: Each example $(q, a, p)$ is annotated by 3 independent raters. We report the agreement with consensus which measures the proportion of examples that agrees with the majority vote. We report statistics on the in-language attribution scenario (S1) and in the in-English attribution scenario (S2). We also report in the ratings collected in (S1) and (S2) ((S1)≠(S2)) and the disagreement on the portion of examples that have been translated from English ((S1)≠(S2) Tr.).

---

and passage triplet $(q, a, p)$, attribution evaluation measures the portion of answers $a$ attributed to $p$. In contrast, Exact-Match requires a gold answer $\tilde{a}$ to compare to the predicted answer $a$.

In Section 4.2, we show how Exact-Match differs from attribution. We show that some correct answers according to exact-match are not attributed to any retrieved passage, while some non-exactly-matching answers are legitimate answers attributed to reference passages.

## 3 The XOR-AttriQA Dataset

To the best of our knowledge, our work is the first to study the problem of attribution in the cross-lingual setting. To make this study feasible, we collect the first multilingual attribution dataset. We use the attribution evaluation framework defined by Rashkin et al. (2021). We hire Bengali, Finnish, Japanese, Russian, and Telugu-speaking raters for the in-language scenario (S1) and English-speaking raters for the in-English scenario (S2). Our analysis is based on the XOR-TyDiQA dataset (Asai et al., 2021a) in Bengali, Finnish, Japanese, Russian and Telugu. To limit cost, we randomly sample about 50% of the validation set except for Bengali and Telugu (S1) annotations for which we take the entire set. We retrieve the passages and predict the answers using the CORA system. We only evaluate the passages that are fed to the generator. For each *(query, answer, passage)* triplet, we ask three raters to answer "*Is the answer attributed to the passage?*".[5] To ensure the quality of the data collected

---

[5]Before this step, we ensure that the *(query, answer)* pair is fully interpretable by the rater. We point to (Rashkin et al.,

| % Attributable predictions (AIS) of EM | | | | |
|---|---|---|---|---|
| **BN** | **FI** | **JA** | **RU** | **TE** |
| 67.3 | 80.4 | 53.1 | 67.5 | 93.1 |

Table 2: % of answers exactly-matching the gold answer attributed to at least one passage fed to the MGEN.

we report in Table 1 the inter-annotator agreement (IAA). The agreement is above 90% for both the in-language scenario and the In-English scenario for all languages except Japanese. Appendix B provides more detail on the annotation process as well as the agreement with expert annotations on a small sample of the data (cf. Table 9). For each example, we assign the attribution label based on the majority vote of the three raters.

**In-English vs. In-Language Attribution** As reported in Table 1, the inter-annotator agreement observed is similar whether we collect the data in the in-English scenario (S1) compared to the in-language scenario (S2). The only large differences are observed for Telugu, for which the IAA is 8 points above when we collect the data in Telugu. In consequence, we will use the annotation from the in-language scenario (S1) as the gold labels to evaluate all our models (cf. 4 and 5). Indeed, (S1) evaluates what a potential user may experience. So given the fact that (S1) is as good (or better) as (S2) concerning data quality, we decide to select the data from (S1).

**Impact of Translation on Attribution Assessment** Both scenarios require translating a portion of the data automatically (cf. Fig. 1). We hypothesize that translating passages from English to the user language may lead, in some cases, to losing attribution. Indeed, assuming that a passage in English supports the answer, we can easily imagine cases in which translation errors could cause the translated passage not to carry the information that supports the answer. However, the disagreement between the annotation in (S1) and (S2) is not higher when we look at passages that have been translated compared to passages that have not for 4/5 languages (cf. comparison between row (S1)≠(S2) and row (S1)≠(S2) TR.) as reported in Table 1). In addition, after manually reviewing the data, we do not find any cases where translation errors cause disagreement.

2021) for an exhaustive definition of interpretability.

**Raters' Demographic and Cultural Background** Even though translating passages does not lead to higher disagreement compared to the original passages, we do observe disagreement between (S1) and (S2) (between 4.1% and and 10.5% as reported in Table 1). We partially explain the disagreement by the demographic and cultural context of the raters. For instance, English speakers rated the example <Query: *How many countries are there in the United States of America?* Answer: *50>* as attributed to the passage *"The United States of America (USA), commonly known as the United States (U.S. or US) or America, is a country composed of 50 states, a federal district, five major self-governing territories, and various possessions."* while Telugu speakers did not do the same for the example translated into Telugu. We hypothesize that a familiarity with the USA and the concept of states made the raters understand the question more loosely and accept the "50 states" mention as supportive of the answer. We leave for future work the careful quantification of this phenomenon.

## 4 Attribution Evaluation of CORA

Based on our newly collected XOR-AttriQA dataset, we now evaluate the attribution level of CORA.

### 4.1 Lack of Attribution of XORQA Predictions

We start by focusing on the subset of answers that match the gold reference based on Exact Match (EM). We hypothesize that these answers are attributable in most cases and that non-attributable answers should be the exception, not the rule. Indeed, by design, CORA uses retrieved passages to generate answers. Intuitively, it is hard to conceive how the model could generate a correct answer without supporting passages. However, it is known that language models "memorize" knowledge in their parameters (Petroni et al., 2019; Roberts et al., 2020), which could enable this ability.

We report in Table 2 the proportion of answers that match the gold reference and that are attributable to a retrieved passage. To our surprise, we find a very large number of non-attributable answers. For Japanese, only 53.1% of the answers are *attributed* to at least one passage.

We provide examples of non-attributed exactly-matching answers in Figure 2. We find that these non-attributed answers exactly-matching the refer-

| | BN | | | FI | | | JA | | | RU | | | TE | | |
|---|---|---|---|---|---|---|---|---|---|---|---|---|---|---|---|
| | AIS | of EM | non-EM | AIS | of EM | non-EM | AIS | of EM | non-EM | AIS | of EM | non-EM | AIS | of EM | non-EM |
| ANY | 27.9/45.6 | 41.8/67.3 | 25.2/40.5 | 38.7/50.9 | 67.9/80.4 | 27.1/39.6 | 11.8/37.3 | 22.4/53.1 | 8.2/23.7 | 27.5/40.9 | 45.0/67.5 | 24.8/37.9 | 23.3/31.7 | 72.4/93.1 | 13.2/19.2 |
| LANG | 25.0/40.2 | 41.8/65.5 | 22.3/36.1 | 36.5/46.0 | 64.3/75.0 | 25.7/34.7 | 11.8/34.8 | 22.4/51.0 | 8.2/20.6 | 26.4/39.8 | 45.0/67.5 | 23.4/36.6 | 22.9/31.4 | 69.0/93.1 | 13.2/18.9 |
| EN | 2.3/3.3 | 0.0/0.0 | 2.6/3.8 | 1.0/4.0 | 3.6/5.3 | 0.0/3.5 | 0.0/2.0 | 0.0/0.0 | 0.0/3.1 | 1.1/1.1 | 0.0/0.0 | 1.4/1.4 | 0.3/0.3 | 1.7/0.0 | 0.0/0.3 |

Table 3: % of attributed answers to the TOP-1/ALL passages fed to MGEN. We report **AIS** (cf. sec. 2.4) on XOR-TyDiQA validation split. **of EM** corresponds to the % of attributable answers among the Exact-Matched answers. **non-EM** corresponds to the % of attributable answers among the non-Exact-Matched answers (i.e. that differ from the gold answer). We report the attribution-level considering passages in any languages (row ANY), only the English passages as our candidates (row EN), only the in-language passages (row LANG.) as our candidates.

**Query**: カール・マルクスは歴史学派？ *Was Karl Marx a historian?* **Answer**: Yes  **Gold Answer**: Yes
**Passage**:
マルクス主義（マルクスしゅぎ、）とは、カール・マルクスとフリードリヒ・エンゲルスによって展開された思想をベースとして確立された社会主義思想体系の一つである。しばしば科学的社会主義（かがくてきしゃかいしゅぎ）とも言われる。マルクス主義は、資本を社会の共有財産に変えることによって、労働者が資本を増殖するためだけに生きるという賃労働の悲惨な性質を廃止
*Marxism is one of the socialist thought systems established based on the ideas developed by Karl Marx and Friedrich Engels. It is often called scientific socialism. By turning capital into the common property of society, Marxism abolishes the disastrous nature of wage labour, in which workers live only to multiply capital.*

**Query:** মারমা জনগোষ্ঠীর মাতৃ ভাষার নাম কী ? *What is the name of the mother tongue of the Marma people?*
**Answer**: বর্মী *Burmese*  **Gold Answer**: বর্মী *Burmese*
**Passage**:
বর্মী ভাষা বা মায়ানমার ভাষা চীনা-তিব্বতী ভাষা পরিবারের তিব্বতী-বর্মী শাখার লোলো-বর্মী উপশাখার একটি ভাষা। বর্মীভাষী জনগণ ঠিক কবে মায়ানমারে এসেছিল, তা বলা যায় না। তবে বর্মী ভাষায় লেখা সবচেয়ে প্রাচীন ধর্মীয় লেখাগুলি খ্রিষ্টীয় ১০ম শতক পর্যন্ত পুরনো। ধারণা করা হয় মধ্য মায়ানমারের নিম্ন উপত্যকাভূমিতে প্রচলিত একটি উপভাষা থেকে আদর্শ বর্মী ভাষার উৎপত্তি ঘটে। বর্তমানের মায়ানমারের অধিকাংশ জনগণ এই বর্মী ভাষার কোন না কোন আঞ্চলিক উপভাষায় কথা বলেন। বর্মী ভাষার প্রথমে পালি ও পরে মোন ভাষার (১২শ -১৩শ শতক) প্রভাব পড়ে। এরপর ১৬শ থেকে ১৯শ শতক পর্যন্ত ভাষাটি বিভিন্ন ইউরোপীয় ভাষা যেমন পর্তুগিজ, ওলন্দাজ, ইংরেজি ও ফরাসি ভাষার সংস্পর্শে আসে। এই
*The Burmese language or the language of Myanmar is a language of the Lolo-Burmese sub-branch of the Tibeto-Burmese branch of the Sino-Tibetan language family. Exactly when the Burmese people came to Myanmar cannot be said. However, the oldest religious texts written in Burmese date back to the 10th century AD. Standard Burmese is thought to have originated from a dialect of the lower valleys of central Myanmar. Most people in present-day Myanmar speak some regional dialect of this Burmese language. Burmese was influenced first by Pali and then by Mon (12th-13th centuries). Then, from the 16th to the 19th century, the language came into contact with various European languages, such as Portuguese, Dutch, English and French. this*

Figure 2: Examples of CORA correct answers not attributed to any passage. We illustrate how the model can be guided to generate correct answers which are not fully supported by the passage.

ence are of various types. Some of these answers seem to be random guesses from the generator that happen to be matching the gold reference regardless of the quality of the retrieved passages. This is usually the case for Yes/No answers. Some answers are correct and seem to be using the information in the passage provided to the generator. However, in most cases, the information provided in the passage is incomplete to support the answer. This is the case for the second example in Figure 2: "What is the name of the mother tongue of the Marma people?" was answered with "Burmese". While the passage contains relevant information about the Burmese language, it does not draw a connection with the "Marma people" mentioned in the question.

These results show—without ambiguity—that ORQA systems, even when they generate correct answers, do not always provide a relevant source passage to support the generated answer. In other words, this means that for a significant portion of answers, ORQA systems are right but without any evidence—they are right for the wrong reasons.

## 4.2 Analysis of CORA's Attribution Level

We now analyze the attribution level of all answers predicted by CORA, not only the correct ones. We report in Table 3 the attribution level of CORA. Depending on the language, between 11.8% (for JA) and 38.7% (for FI) of answers are attributed to the TOP-1 passage retrieved by MDPR. In addition, for all languages, we find that between 31.7–50.9% of answers are attributed to at least one passage in the ones provided to the generator (ALL).

**Impact of Cross-Language Attribution** One of the key ingredients of the performance of CORA is its ability to use passages cross-lingually (mainly in English) (Asai et al., 2021b). We now look at how often the generated answers are attributable to these cross-lingually retrieved passages. We find that between 0.3% and 4.0% of answers in Telugu and Finnish respectively can be attributed to an English passage (while not being attributed to any passage in the same language as the query; cf. EN row in Table 3).

**Attribution vs. Exact-Match** In Section 4.1, we found that a large portion of answers exactly matching the gold reference are not attributable. We now look at the answers that are not exactly matching the reference (cf. column non-EM in Table 3). We hypothesize that attribution can potentially complement string-matching metrics and find answers that otherwise would be considered incorrect. In Telugu, we find that 13.2% of such answers are attributed to the TOP-1 passage. We provide such examples in the Appendix in Figure 3. Some answers are semantically equivalent to the gold reference but are spelled differently or employ different measuring units (e.g., "crore" used in Telugu vs. "ten million"). Some answers are semantically different for the gold reference but are attributable to a passage (e.g., *the liver* as the largest organ).

## 5 Attribution Detection for XORQA

So far, we found that state-of-the-art cross-lingual question answering systems lack attribution. We showed that a large portion of answers are not attributed to any passages, by collecting a large collection of attribution data in five languages.

However, collecting attribution data is costly and time consuming. In a deployment setting, it would simply be infeasible to annotate every *(query, answer, passage)* triplet. In order to address this issue, we explore automatic attribution detection techniques. We build upon previous work on grounding and factual consistency in English (Honovich et al., 2022). We also experiment with PaLM 2 (Anil et al., 2023) a new state-of-the-art multilingual large language model (LLM) in few-shot and scarce data (250 examples) settings.

### 5.1 Attribution Detection Models

Given a query $q$, a short answer $a$ and a passage candidate $p$, we frame attribution detection as a binary classification task:

$$(q, a, p) \rightarrow \widehat{ais} \qquad (2)$$

with $\widehat{ais} \in \{0, 1\}$. 1 corresponds to the attributed class (i.e., the answer is attributed to the passage) and 0 corresponds to the non-attributed class. We note that query and answers are always in the same language (in Bengali, Finnish, Japanese, Russian or Telugu), while the passage may be in a different language (mainly English). Following Honovich et al. (2022), we model this task by prompting the

models as follows: *premise: "$p" hypothesis: the answer to the question "$q" is "$a"* where $p$, $q$, and $a$ are inserted appropriately.

**MT5-QA** We use the training splits of the TyDi QA dataset (Clark et al., 2020) to train the attribution detection model. We employ the query, passage, answer triplets from TyDi QA as our attributed examples (our positive class). For non-attributed examples, we mine negative passages as follows: given a query, we start with the entire Wikipedia document from TyDi QA that answers the query. We sample from this document 10 passages that are different from the positive passage (i.e. the passage that answers the query). This technique provides strong negative passages by providing passages that are topically closely related to the positive passage but that do not answer the question. It was used successfully by Garg et al. (2020). We fine-tune mT5-XXL (Xue et al., 2021) on the concatenation of the training data in English, Bengali, Finnish, Japanese, Russian and Telugu.

**(M)T5-NLI** Following Honovich et al. (2022) who found that NLI-fine-tuned T5 is accurate for factual consistency detection, we experiment with several English and multilingual NLI models. Similar to Bohnet et al. (2022), we make use of the best English NLI model from Honovich et al. (2022), a T5-11B model fine-tuned on a mixture of natural language inference datasets, fact verification, and paraphrase detection datasets.[6] We experiment with it in the translate-test setting (noted T5-NLI TRANSLATE-TEST) for which we translate the queries, passages, and answers to English.[7] To model attribution detection in multiple languages, we fine-tuned the mT5-XXL model (Xue et al., 2021) on translations of the mixture of the NLI datasets to the non-English languages (noted MT5-NLI TRANSLATE-TRAIN). To better model the portion of passages in a language different from the query, we also fine-tune the model by adding examples for which only the hypothesis has been translated while the premise is kept in English (noted MT5-NLI X-TRANSLATE-TRAIN).

**PALM 2 FEW SHOT** To avoid costly fine-tuning, we experiment with in-context learning using the

---

[6]T5 is fine-tuned on the concatenation of the MNLI (Nangia et al., 2017), SNLI (Bowman et al., 2015), FEVER (Thorne et al., 2018), PAWS (Zhang et al., 2019), SciTaiL (Khot et al., 2018) and VitaminC (Schuster et al., 2021) datasets.

[7]Except for the portion of the passages retrieved in English that do not need translation.

| Model | Tuning Data (#) | Inference | BN | FI | JA | RU | TE |
|---|---|---|---|---|---|---|---|
| STRING-MATCH | Ø | IN-EN | 83.9 / 72.0 | 85.4 / 75.8 | 91.9 / 71.7 | 86.8 / 74.5 | 87.0 / 84.5 |
| STRING-MATCH | Ø | IN-LANG | 87.1 / 78.5 | 85.6 / 78.3 | 90.4 / 77.3 | 87.5 / 80.0 | 88.6 / 88.3 |
| MT5-QA-TRANSLATE-TEST | TyDiQA (∼100k) | IN-EN | 88.0 / 91.7 | 83.5 / 91.5 | 90.2 / 90.2 | 86.6 / 92.4 | 87.8 / 94.2 |
| MT5-QA | TyDiQA (∼100k) | IN-LANG | 89.4 / 92.0 | 88.3 / 92.2 | 91.5 / 92.9 | 91.0 / 94.7 | 92.4 / 96.8 |
| T5-NLI-TRANSLATE-TEST | NLI (∼1M) | IN-EN | 91.8 / 95.2 | 91.2 / 96.0 | 95.1 / 95.7 | 92.4 / 96.0 | 94.2 / 95.1 |
| MT5-NLI TRANSLATE-TRAIN | NLI (∼1M) | IN-LANG | 91.1 / 93.8 | 90.4 / 94.6 | 93.0 / 93.8 | 92.9 / 96.1 | 93.8 / 96.0 |
| MT5, FINE-TUNED | Attribution (∼100) | IN-LANG | 81.9 / 71.7 | 80.9 / 69.8 | 94.5 / 64.8 | 87.1 / 67.1 | 88.7 / 78.7 |
| PALM 2, FINE-TUNED | Attribution (∼100) | IN-LANG | **92.3** / 95.0 | **92.6** / **97.2** | **96.4** / 95.9 | **94.5** / **98.2** | **94.8** / 96.8 |
| PALM 2, LoRA-TUNED | Attribution (∼100) | IN-LANG | 91.5 / **96.0** | 88.3 / 96.4 | 94.7 / **97.0** | 93.7 / 97.8 | 93.7 / **96.8** |
| PALM 2 4-SHOT | Attribution (∼4) | IN-LANG | 91.5 / 87.3 | 87.4 / 82.1 | 92.0 / 85.6 | 90.5 / 78.7 | 90.6 / 77.3 |
| PALM 2 4-SHOT w/ CoT PROMPT | Attribution (∼4) | IN-LANG | 83.7 / 86.8 | 78.8 / 85.3 | 71.7 / 80.4 | 81.9 / 88.0 | 84.7 / 88.6 |

Table 4: Performance of Attribution detection models. We report the Accuracy / ROC AUC scores on the XOR-AttriQA dataset. The Accuracy is computed with the probability threshold that maximizes it on an independent set. For each language, the best Accuracy / ROC AUC scores are bolded and the second best scores are underlined.

PaLM 2 large language model (Anil et al., 2023). We use the Small version and evaluate the model after prompting the model with 4-shots with and without chain-of-thought prompting (Wei et al., 2022). Each language is evaluated with its own prompts, and two negative examples and two positive examples are sampled for each language. For each pair, one passage is chosen to be in-language while the other is chosen to be in-English. Chain-of-thought is done by manually writing a rationale that explains the attribution (or lack of attribution) of a given answer in English.

**MT5 / PALM 2 - ATTRIBUTION** Finally, we experiment with fine-tuning directly on a small sample of the attribution data we collected. We sample 250 examples in the 5 languages and fine-tune mT5-XXL (Xue et al., 2021) and PaLM 2 Small (Anil et al., 2023). For mT5 we fine-tune the entire model, while for PaLM 2, we both fine-tune on the whole dataset and also fine-tune with Low-Rank Adaptation (LoRA) (Hu et al., 2021) to avoid overfitting and reduce fine-tuning cost. For these experiments, we use the same constant learning rate of 0.0001, dropout rate (Srivastava et al., 2014) of 0.1, and batch size of 128 for tuning both mT5 and PaLM 2. For fine-tuning with LoRA, we used a learning rate of 0.00005 and tuned the model over ranks in $\{4, 16, 64, 256\}$. For all models, we used the validation set for checkpoint selection.

**STRING-MATCH** We define a simple baseline. For answers that are not "Yes"/"No", if the string $a$ is included in the passage $p$, we predict 1, otherwise 0. This means that we consider the answer to be attributed to the passage if it is included in it. For Yes/No answers, we predict 0 (the majority class). We also use it after translating the query, answer and passage to English.[8]

## 5.2 Results

We report the accuracy and ROC-AUC (Flach et al., 2011) scores in Table 4. We compute the prediction with a decision threshold tuned on an independent validation dataset on which we measure the accuracy of the model. PaLM 2 outperforms all the other models despite being fine-tuned on a very small sample of data, which is encouraging as it shows we can leverage LLMs for cross-lingual attribution with very little annotated data that is expensive to produce. We also found few-shot performance to also have strong results, that could probably be improved with more shots and leveraging larger LLMs. NLI fine-tuned models outperform MT5-QA and STRING-MATCH for all the languages in the translate-test setting. Finally, we report in Table 5 the portion of attributed answers not matching the gold references that the best AIS detection model accurately predicts. We find that PaLM 2 accurately predicts more than 86% of these answers (between 86.3 and 96.4% depending on the language). This shows the potential of using attribution detection to expand the space of legitimate answers beyond relying only on string-matching metrics.

| | BN | FI | JA | RU | TE |
|---|---|---|---|---|---|
| Acc. | 88.2 | 94.5 | 86.3 | 96.4 | 91.6 |

Table 5: % of attributed non-EM examples that are accurately detected by PALM 2.

| Lang. | TOP-1 | ALL | T5-NLI reranked |
|---|---|---|---|
| BN | 27.9 | 45.6 | 39.2 (+40.4%) |
| FI | 38.7 | 50.9 | 46.0 (+19.0%) |
| JA | 11.8 | 37.3 | 29.1 (+146.2%) |
| RU | 27.5 | 40.9 | 39.6 (+43.9%) |
| TE | 23.3 | 31.7 | 30.3 (+30.0%) |
| Avg. | 25.8 | 41.3 | 36.8 (+55.9) |

Table 6: % of attributed answers based on the top-1 MDPR-retrieved passage, ALL the passages retrieved fed to the generator, and the TOP-1 reranked passage (T5-NLI reranked) with T5-NLI-TRANSLATE-TEST, our best NLI fine-tuned model.

## 5.3  NLI Model for Reranking

Using our best T5-based attribution detection model (T5-NLI TRANSLATE-TEST), we now come back to our original goal of improving the attribution level of our cross-lingual question answering system. We leave for future work the use of PaLM 2 for reranking.

Given our pool of candidate passages, we use our attribution detection model as a reranker and select the passage which is the most likely to attribute the answer according to the model. We report the reranking attribution score in Table 6. We find that our NLI model can accurately rerank the passages. For instance for Telugu, we are able to increase the top-1 performance from 23.3 to 31.7, an improvement of +30.0%. Across all the languages, reranking with our NLI model leads to am average relative increase of 55.9% across the 5 languages.

## 6  Discussion and Future Directions

Language model-based NLP systems are making fast and continuous progress in generating fluent and helpful text. Despite this progress, these models still make a lot of factual errors, specifically in languages different from English. Attribution is the most promising approach in addressing this issue (Rashkin et al., 2021). Our work finds that even the best XORQA system predictions lack attribution. These results can be explained by the tendency of these models to memorize facts (Roberts et al., 2020) and to hallucinate answers (Ji et al., 2023), which are in some cases correct. This shows that we need to make progress in detecting and selecting attributed sources that support the generated answer of cross-lingual QA systems. In this work, we proposed to use a large language model (PaLM 2) and natural language inference models to detect and

---

[8]Using translation ensures that everything is in the same language potentially improving the string-matching accuracy.

rerank passages to improve the attribution-level of a state-of-the-art XORQA system.

Our result points to two critical research directions to further make progress in information-seeking QA. First, we observed that in some languages (e.g., Telugu), cross-lingual passages contribute very moderately to the attribution level. This shows that more progress is needed in cross-lingual retriever systems. Second, we showed that string-matching metrics based on gold references are inherently imperfect evaluation methods for QA and showed that PaLM 2 can be used to detect relevant attributed passages accurately with only small amounts of training data. This means that large language models-based attribution detection can potentially be used as evaluation metrics for QA in multiple languages. Further work is needed to design robust LLM-based metrics for cross-lingual information-seeking QA.

## 7  Conclusion

By ensuring that the model predictions are supported by human-written text, attribution is one of the most promising ways to deploy NLP systems safely. In this work, we introduced and released the XOR-AttriQA dataset that includes approximately 10,000 examples in Bengali, Finnish, Japanese, Russian and Telugu. Thanks to XOR-AttriQA, we observe that state-of-the-art QA systems lack attribution in the cross-lingual setting. We showed that PaLM 2 and NLI models are promising methods to detect attributed passages in 5 typologically diverse languages for information-seeking QA. Having provided evidence for the lack of attribution in academic generative QA system, built tooling to detect and mitigate these issues, and releasing our collected attribution data in 5 languages, we hope to enable trustworthy cross-lingual QA systems to meet the information needs of people around the world.

## Limitations

Our work focused on evaluating and improving the attribution level of a state-of-the-art XORQA pipeline. Given recent progress, LLMs are now increasingly used in a closed-book setting (Roberts et al., 2020) for question answering, i.e., without relying on any retrieved passages to answer a question. Attributing the generations of these models is therefore becoming critical. In addition to improving the attribution level of open-retrieval question

answering pipeline, we hope XOR-AttriQA and the attribution detection experiments we presented will also be used to design attribution detection models for closed-book QA systems.

## Contributions

In this section, we provide more detail about the contributions of each author.

### General Overview

**Primary Contributors** Benjamin Muller, John Wieting

**Major Contributors** Jonathan H. Clark, Tom Kwiatkowski, Sebastian Ruder, Livio Baldini Soares

**Supporting Contributors** Roee Aharoni, Jonathan Herzig, Xinyi Wang

### By Contribution

**Advising** John Wieting, Jonathan H. Clark, Tom Kwiatkowski, Sebastian Ruder, Livio Baldini Soares, Roee Aharoni, Jonathan Herzig

**Code** Benjamin Muller, John Wieting, Livio Baldini Soares, Roee Aharoni, Jonathan Herzig

**Data Collection** Benjamin Muller, John Wieting

**Experiments** Benjamin Muller, John Wieting

**Modeling** Benjamin Muller, John Wieting, Xinyi Wang, Roee Aharoni, Jonathan Herzig

**Project Design** Jonathan H. Clark, John Wieting, Benjamin Muller, Tom Kwiatkowski, Sebastian Ruder

**Project Initiation** Jonathan H. Clark, Tom Kwiatkowski

## Acknowledgements

We thank the raters involved in the data collection process for their work. In addition, we want to thank Michael Collins, Dipanjan Das, Vitaly Nikolaev, Jason Riesa, and Pat Verga for the valuable discussion and feedback they provided on this project.

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

# Appendices accompanying "Evaluating and Modeling Attribution for Cross-Lingual Question Answering"

## A  System and Data

### A.1  Codebase

To run inference with CORA (Asai et al., 2021b), we used the original codebase released by the authors available at https://github.com/AkariAsai/CORA. To build attribution detection models with T5, we used the original checkpoints from (Xue et al., 2021) fine-tuned using the t5x library available at https://github.com/google-research/t5x.

### A.2  XOR-TyDiQA

| Language | yes/no | short spans | all |
|---|---|---|---|
| BN | 35.3 | 45.8 | 45.6 |
| FI | 45.2 | 52.1 | 50.9 |
| JA | 37.0 | 37.6 | 37.3 |
| RU | 36.0 | 41.2 | 40.9 |
| TE | - | 31.7 | 31.7 |

Table 7: % of CORA's answers that are attributed to at least one passage per answer type.

All our experiments are based on the XOR-TyDiQA dataset (Asai et al., 2021a) available at https://nlp.cs.washington.edu/xorqa/. We focused on Bengali, Finnish, Japanese, Russian and Telugu data. We only used the query and gold answers from XOR-TyDiQA (and ignored the gold passages for which we use a retriever). XOR-TyDiQA answers are of two types: Yes/No answers or short spans extracted from a Wikipedia passages (Clark et al., 2020). We report in Table 7 the difference in attribution between Yes/No and short span answers. We find that for most languages, Yes/No answers are less attributable compared to short answer spans.

### A.3  Languages Distribution of MDPR

| Query lang. / Passage lang. | Language Distribution of Retrieved Passages | | | | |
|---|---|---|---|---|---|
| | BN | FI | JA | RU | TE |
| IN-LANG | 44.9 | 69.2 | 82.8 | 87.3 | 55.8 |
| EN | 49.1 | 22.9 | 15.7 | 10.6 | 40.0 |
| OTHERS LANG. | 5.9 | 7.9 | 1.5 | 2.1 | 4.2 |

Table 8: Language distribution of passages retrieved by a multilingual dense retriever (MDPR from (Asai et al., 2021b)). IN-LANG means the passage is in the same language as the query.

Table 8 shows the language distribution of passages retrieved by MDPR. Most passages are either in the same language as the query or in English.

## B  Data Collection

### B.1  Inter-Annotator Agreement Details

| Lang. | Scenario | Rater Consensus w. Expert Agreement | Individual Rater with Expert Agreement |
|---|---|---|---|
| BN | IN-LANG (S1) | 91.18 | 88.78 |
| BN | IN-EN (S2) | 86.49 | 84.68 |
| FI | IN-LANG (S1) | 96.67 | 93.33 |
| FI | IN-EN | 100.00 | 88.89 |
| JA | IN-LANG (S1) | 93.62 | 91.19 |
| JA | IN-EN | 83.33 | 72.22 |
| RU | IN-LANG | 93.75 | 94.61 |
| RU | IN-EN | 100 | 88.89 |
| TE | IN-LANG (S1) | 95.75 | 88.58 |
| TE | IN-EN (S2) | 91.23 | 85.10 |

Table 9: Attribution annotations quality: we report the agreement between hired raters and expert judgment on a small sample of data.
NB: Expert IN-LANG evaluation is done based on translated queries and answers.

We report in Table 9 the agreement between expert raters and hired raters on a small number of examples. We find that this agreement is above 90% for all languages in the in-language scenario (S1).

### B.2  AIS Score

The AIS data collection framework (Rashkin et al., 2021) consists of two annotation steps. First, the raters are shown a question and answer and asked "Is the answer interpretable to you". If the response is positive, the rater is shown the source passage and asked "Is the answer attributed to the passage". At each step, the rater is asked to answer Yes, or No or to flag the example if it is corrupted. For each question, answer, and passage triplet $(q, a, p)$, we collect the rating of three raters (for each annotation scenarios). These three ratings are aggregated to get a single label $0$ or $1$ for each $(q, a, p)$ triplet with the following criterion:

- We only keep the examples that received at least two ratings. This means that we exclude examples flagged by two raters or more.

- We assign the label "attributable" (1) to the triplet if the example received at least two votes to the question "Is the answer attributed to the passage"; otherwise, we set the label to non-attributable (0).

The number of examples collected is available in Table 1.

## C Examples of Attribution without Exact-Match

---

**Query:** కెన్యా దేశ రాజధాని ఏమిటి? *What is the capital of Kenya?* **Answer** నైరోబి *Nairobi* **Gold answer**: నైరోబి *Nairobi*
**Passage**:

కెన్యా (ఆంగ్లం Republic of Kenya) రిపబ్లిక్ ఆఫ్ కెన్యా, తూర్పు ఆఫ్రికా లోని ఒక దేశం. దీని ఉత్తరాన ఇథియోపియా, ఈశాన్యాన సోమాలియా, దక్షిణాన టాంజానియా దేశాలు గలవు. దీని రాజధాని నైరోబి.
*Kenya (English Republic of Kenya) The Republic of Kenya is a country in East Africa. It is bordered by Ethiopia to the north, Somalia to the northeast and Tanzania to the south. Its capital is Nairobi.*

---

**Query**: రెండవ ప్రపంచయుద్ధంలో సగటుగా ఎంతమంది చనిపోయారు? *How many people died on average in World War II?*

**Answer**: ఆరు కోట్ల *Six crores*

**Gold answer**: 70-85 మిలియన్లు *70-85 millions*
**Passage**:

కోట్లు. ఇందులో పాల్గొన్న దేశాలు ఒక రకమయిన పరిపూర్ణ యుద్ధ పరిస్థితిని ఎదుర్కొన్నాయి (అనగా, సైనిక-పౌర భేదాలు లేకుండా అందుబాటులో ఉన్న వారందరూ ఏదో ఒక రకంగా యుద్ధంలో పాలుపంచుకోవటం). ఆకారణంగా ఆయా దేశాల ఆర్థిక, పారిశ్రామిక, సాంకేతిక వనరులన్నిటినీ యుద్ధ ప్రయోజనాలకోసమే వాడవలసి వచ్చింది. సుమారు ఆరు కోట్లమంది మృతికి కారణమయిన ఈ యుద్ధం ప్రపంచ చరిత్రలోనే అత్యంత రక్త సిక్తమయినదిగా పేరొందింది. రెండవ ప్రపంచ యుద్ధంలో మరణించిన వారిలో మూడింట రెండు వంతులు సాధారణ పౌరులేనని ఒక అంచనా. వీరిలో సుమారు ఒక కోటిమంది వరకూ తూర్పు ఐరోపాలోనూ సోవియెట్ యూనియన్ లోనూ నాజీ జర్మనీ జరిపిన యూదు జాతి నిర్మూలన కార్యక్రమంలో ప్రాణాలు పోగొట్టుకున్నారు (దీనికి హోలోకాస్ట్ అని పేరు). ప్రపంచ వ్యాప్తంగా ఈ యుద్ధం కలిగించిన ఆర్థిక నష్టం సుమారు పది లక్షల కోట్ల అమెరికన్ డాలర్లు (1944 నాటి డాలరు విలువ ప్రకారం) ఉంటుందని
*crores. The countries involved faced a kind of perfect war situation (ie, all available, regardless of military-civilian distinctions, were involved in the war in some way). As a result, all the economic, industrial and technological resources of the respective countries had to be used for war purposes. This war is known as the bloodiest in the history of the world, which caused the death of about six crore people.*

---

**Query**: మానవ శరీరంలోని అతి పెద్ద అవయవం ఏది ? *Which is the largest organ in the human body?*
**Answer**: చర్మము *the skin* **Gold answer**: కాలేయము *the liver*
**Passage**:

చర్మము (Skin) మన శరీరంలో అతిపెద్ద అవయవము. దీనిలో మూడు ముఖ్యమైన పొరలుంటాయి. చర్మము శరీరమంతా కప్పి లోపలి భాగాల్ని రక్షిస్తుంది. నవరంధ్రాలవద్ద చర్మం లోపిస్తుంది. ఇది వివిధ రంగులలో ఉంటుంది. చర్మానికి సంబంధించిన విజ్ఞాన శాస్త్రాన్ని 'డెర్మటాలజీ' అంటారు. చర్మంలో ముఖ్యంగా బాహ్యచర్మం, అంతశ్చర్మం అనే రెండు పొరలుంటాయి. బాహ్యచర్మం బహిస్స్పచం నుంచి ఏర్పడుతుంది. రోమాలు, స్వేద గ్రంధులు బాహ్యచర్మానికి చెందినవి. గోళ్ళు కూడా దీనినుంచే ఏర్పడతాయి. ఆఫ్రికా దేశీయులు నల్లగా ఉంటారు. ఉత్తర ఐరోపా దేశీయులు తెల్లగా ఉంటారు. ఆసియా మరికొన్ని ప్రాంతాల ప్రజలు వీరిరువురి మధ్యలో ఉంటారు. ఈ వర్ణభేదాలకు కారణం చర్మంలోని 'మెలనిన్' అనే రంగుపదార్థం.low melanin is called albuns.
*Skin is the largest organ in our body. It has three important layers. The skin covers the entire body and protects the internal parts. Skin is lacking at the pores. It comes in different colors. The science of skin is called 'Dermatology'. The skin mainly has two layers namely epidermis and dermis. The epidermis is formed from the epidermis. Hairs and sweat glands belong to the epidermis. Nails are also formed from it. Africans are black. Northern Europeans are white. The people of some other parts of Asia are in between the two. The cause of these color differences is the pigment called 'melanin' in the skin. low melanin is called*

---

Figure 3: Examples of attributed answers that are not matching the gold reference (noted non-EM in Table 3). We display a single passage fed to the generator, to which the answer is attributed.