# OpenReview forum: "Evaluating and Modeling Attribution for Cross-Lingual Question Answering"
_EMNLP/2023/Conference — EMNLP 2023 Main_

### Official Review · Reviewer_zM5q · 2023-08-04

**Soundness:** 4

**Excitement:**

4: Strong: This paper deepens the understanding of some phenomenon or lowers the barriers to an existing research direction.

**Paper Topic And Main Contributions:**

The authors make the following four contributions:
1. Their work is the first to study attribution for question answering in a cross-lingual framework. They define two attribution scenarios, in-language attribution and in-English attribution and annotate about 10,000 examples in five languages;
1. using this data they evaluate the attribution of CORA, a state-of-the-art cross-lingual QA system. They show that a large portion (7%-47%, depending on the language) of the answers are not attributable neither to in-language passages nor to cross-language ones;
1. they show that PaLM 2 and NLI models can accurately detect attribution in the cross-lingual setting, significantly outperforming all other baselines and reaching above 90% accuracy for all 5 languages; and
1. using their attribution detection model as a reranker, they show that the model reach an average of +55% in attribution compared to a model with no reranking.

**Reasons To Accept:**

Attribution is the most promising approach in addressing factual errors. This paper finds that even the best XORQA system predictions lack attribution. This paper proposed to use a large language model and natural language inference models to detect and rerank passages to improve the attribution-level of a state-of-the-art XORQA system.

**Reasons To Reject:**

I don't see a big reason to reject this paper.

**Reproducibility:**

3: Could reproduce the results with some difficulty. The settings of parameters are underspecified or subjectively determined; the training/evaluation data are not widely available.

**Reviewer Confidence:**

2: Willing to defend my evaluation, but it is fairly likely that I missed some details, didn't understand some central points, or can't be sure about the novelty of the work.

---

> ### Author Rebuttal · Authors · 2023-08-29
>
> Thank you for your review. We share your excitement. We agree that attribution is a very promising approach to addressing factual errors, and we hope our work makes a meaningful contribution to building better attributed systems.

---

### Official Review · Reviewer_TzMb · 2023-08-05

**Soundness:** 4

**Excitement:**

3: Ambivalent: It has merits (e.g., it reports state-of-the-art results, the idea is nice), but there are key weaknesses (e.g., it describes incremental work), and it can significantly benefit from another round of revision. However, I won't object to accepting it if my co-reviewers champion it.

**Paper Topic And Main Contributions:**

The paper is devoted to a novel setup in cross-lingual question-answering, namely attribution evaluation. The authors present also a dataset for evidence evaluation and the evaluation results.

**Reasons To Accept:**

A novel dataset in a novel setup of cross-lingual question answering.

**Reasons To Reject:**

The evaluation of attribution has been done for only one model, although being popular but not the best in current time.

**Reproducibility:**

4: Could mostly reproduce the results, but there may be some variation because of sample variance or minor variations in their interpretation of the protocol or method.

**Reviewer Confidence:**

4: Quite sure. I tried to check the important points carefully. It's unlikely, though conceivable, that I missed something that should affect my ratings.

---

> ### Author Rebuttal · Authors · 2023-08-29
>
> Thank you for your overall positive review.
>
> Indeed, our paper only displays the evaluation of a single cross-lingual open-retrieval pipeline (CORA). We managed this project on a budget constrained setting for data collection, and even then we were able to gather about 10,000 data points annotated 3 ways. We, therefore, prioritized both the language-scaling aspect (so we could cover five typologically diverse languages) and evaluating two attribution settings (in-language and in-English attribution) instead of evaluating more models. However, we note that CORA is a standard open-retrieval pipeline, and our approach handles any (query, passage, answer) and is not restricted to CORA outputs. Therefore, we believe our findings on improving attribution will generalize to other pipelines.

---

### Official Review · Reviewer_PwG7 · 2023-08-11

**Typos Grammar Style And Presentation Improvements:** Missing % in the last row of Table 6
**Soundness:** 3

**Excitement:**

4: Strong: This paper deepens the understanding of some phenomenon or lowers the barriers to an existing research direction.

**Missing References:**

Shen et al., 2023: xPQA: Cross-Lingual Product Question Answering in 12 Languages

**Paper Topic And Main Contributions:**

This paper studies attribution of QA in a cross-lingual setting. It introduces two attribution scenarios, in-language and in-English attribution. To assess  existing cross-lingual QA models, they collect 10000 human annotations in 5 languages. They show that in the existing model, a large portion of the answers generated are not attributable. The paper also introduces attribution classifiers by finetuning a small sample on PALM 2 or finetuned T5 variants. Using such classifiers to help rerank the passages, they are above to further improve original system performance.

**Questions For The Authors:**

a. Though palm 2 is only finetuned on a few samples, it is still very costly for a binary classification model. Is there any intuition on why in–context learning does not work well for the attribution task?

b. Why does T5-NLI perform better than mT5-NLI in most cases?


**Reasons To Accept:**

The paper is well written and easy to read. It introduces attribution in cross-lingual QA.  In addition to presenting existing attribution problems in the current system, the paper also collects new annotations, proposes two methods for attribution classification and could further improve QA performance.

**Reasons To Reject:**

- It’s great to see the paper tries to improve explainability in QA systems by attributing answers to retrieved documents. However, framing attribution detection as a binary classification problem does not provide further clarity compared to the ranking provided by the retriever.
- Domain is limited to wikipedia articles. There’s a lack of discussion about how this work could be extended to other domains.
- Comparison between mT5–QA and mT5-NLI. The amount of data used to finetune these two variants are significantly different. I’m wondering if adding more QA data to the former, would that bring the gap closer? Or training on a combination of QA and NLI data would further boost the performance?
- There’s no mention of the compute cost and the training details.
- The authors appear to have not invested much thought into the limitation section, as it does not discuss the paper's own limitations.l


**Reproducibility:**

2: Would be hard pressed to reproduce the results. The contribution depends on data that are simply not available outside the author's institution or consortium; not enough details are provided.

**Reviewer Confidence:**

4: Quite sure. I tried to check the important points carefully. It's unlikely, though conceivable, that I missed something that should affect my ratings.

---

> ### Author Rebuttal · Authors · 2023-08-29
>
> Thank you for your insightful review.
>
> We want to reaffirm some distinctions between our attribution detection models and standard retrievers. First, the retriever is very inaccurate to provide attributed passages. As we reported in Table 6, only 25.8% on average of the top-1 passage provided by the retriever is an attributable source for the predicted answer. This means that a passage selection step is needed to improve this number. Second, our attribution detection system takes as input the (question, passage, answer) triplet, making the attribution detection tailored toward the predicted answer. Using this approach, we find that we are able to improve the attribution level with our attribution detection model by +55.9% on average.
>
> We showcased our system using Wikipedia since that is the domain of the XORQA data and the state-of-the-art CORA model we study. Indeed, further work is needed to showcase our approach on a larger number of domains (e.g., medical, legal text). Our approach is, however, directly applicable to other domains (given an existing XORQA pipeline).  Additionally, attribution detection for general knowledge information-seeking question answering requires a trustworthy corpus of text (as described in "Measuring attribution in natural language generation" by Rashkin et al. (2021)). Due to its scale, multilingual coverage and overall good quality, we believe Wikipedia is the most relevant corpus to attribute answers for information-seeking QA.
>
> The amount of data used for QA is an order of magnitude smaller compared to data to train the NLI models. The number of available multilingual QA training resources is limited. The XOR-TyDiQA was the only resource known to us that includes questions, answers, and full documents allowing us to build an attribution detection training dataset (with both positive and negative passages). We agree that a combination of QA and NLI fine-tuning data will likely improve the performance even more.
>
> Thank you for the feedback about the training details. We will add the computing cost and training hyperparameters to the paper. In summary, we used a batch size of 32 and a constant learning rate of 0.0001. All the training runs were done for up to 50k steps, and the best checkpoint was selected on a validation set that follows the training distribution. In total, the training runs for the NLI attribution classifiers took about 100 hours on a Google Cloud TPUv3 Pod slice with 32 chips. LoRA tuning for PaLM 2 also used a Google Cloud TPUv3 Pod slice with 32 chips and took about 6 hours.
>
> We will expand our Limitations section as per your suggestion (including limitations such as a single domain, using CORA only, training costs, etc.). We agree and thank you for pointing this out.
>
> Using PALM 2 with in-context learning for attribution detection did not compete with the other approaches. Attribution detection requires discriminating between attributed passages and passages that seem supportive of the answer but that are not (see Table 3 for examples). Simple in-context few-shot learning did not deliver good results, significantly below the NLI models, but we believe further prompting exploration could improve the performance. These results, with in-context learning lagging behind fine-tuning, are consistent with the literature as well. For instance, this topic is discussed in Section 4.3 of "XTREME-UP: A User-Centric Scarce-Data Benchmark for Under-Represented Languages" by Ruder et al. (2023) in a similar setting to ours where in-context learning was not as effective as fine-tuning. We include additional results below for in-context learning using 4-shot Chain-of-Thought prompting on PaLM 2 M (using reasoning steps we provide):
>
> | Language | Accuracy | ROC AUC |
> |-----------------|-----------------|-----------------|
> | BN | 79.8 | 86.4 |
> | FI | 77.1 |84.9 |
> | JA | 79.2 | 88.0 |
> | RU | 74.8 | 85.5 |
> | TE | 90.0 | 50.0 |
>
> These in-context learning results are well below that of the fine-tuning experiments in Table 5. We will include these results and discussion in the next version of the paper. Note that for in-context learning for TE, the model became stuck repeating a portion of the prompt.
>
> The difference between T5-NLI and mT5-NLI is a combination of two factors. T5 is, generally speaking, a better language model for English, than mT5 is for other languages. This is mainly due to English having significantly more pretraining data than other languages. Second, mT5-NLI relies on translated fine-tuning data. This may hurt the generalization of the model to attribution detection.
>
> As for reproducibility, our results use publicly available pretained models (with the exception of PaLM-2 which requires an API) and we will be releasing our ~10,000 example dataset to the community as promised in the footnote of the abstract. A preliminary version of the data is [here](https://www.dropbox.com/scl/fi/vyn5jve3mrxgctadays8p/xor-attriqa-v1.0.zip?rlkey=se3xxdi3cr5668zdd1cnlr6ci&dl=0).
>
> Thank you again for the valuable comments; we hope our answers will bring more clarity about our work.

---

### Meta-Review · Area_Chair_zNL9 · 2023-09-17

**Recommendation:** 5

**Metareview:**

The work is the first to study attribution in cross-lingual QA with two scenarios: in-language and in-English attribution. The authors collect multilingual attribution datasets for languages with diverse typologies and analyze the potential confounding factors in the data collection process such as translation quality, interannotators’ agreement, and the annotators’ background. Then, they present the existing attribution problem in the current ORQA systems and perform error analysis. Finally, the authors presented solutions that help cross-lingual attribution for LLMs and improve QA performance substantially. The authors sufficiently addressed reviewer PwG7’s confusion between the retriever and their proposed attribution detection models and justified the study on the Wikipedia domain in their work. The authors also draw upon the results from the paper to answer reviewers’ questions, which suggests that the experimental setup is careful and sound. Note that in the next iteration of the paper, the training costs should be included as the paper works with extremely large LMs such as PaLM 2.

I think the work is timely as the community is working on the trustworthiness and explainability of large language models that are currently widely used for user-facing QA tasks. Agreeing with reviewer PwG7, I think the paper is well-written and easy to read.

---

### Decision · Program_Chairs · 2023-10-07

**Decision:**

Accept-Main

**Comment:**

The work is the first to study attribution in cross-lingual QA with two scenarios: in-language and in-English attribution. The authors collect multilingual attribution datasets for languages with diverse typologies and analyze the potential confounding factors in the data collection process such as translation quality, interannotators’ agreement, and the annotators’ background. Then, they present the existing attribution problem in the current ORQA systems and perform error analysis. Finally, the authors presented solutions that help cross-lingual attribution for LLMs and improve QA performance substantially. The authors sufficiently addressed reviewer PwG7’s confusion between the retriever and their proposed attribution detection models and justified the study on the Wikipedia domain in their work. The authors also draw upon the results from the paper to answer reviewers’ questions, which suggests that the experimental setup is careful and sound. Note that in the next iteration of the paper, the training costs should be included as the paper works with extremely large LMs such as PaLM 2.

I think the work is timely as the community is working on the trustworthiness and explainability of large language models that are currently widely used for user-facing QA tasks. Agreeing with reviewer PwG7, I think the paper is well-written and easy to read.